# Bisphenol A Diglycidyl Ether-Primary Amine Cooligomer-poly(ε-caprolactone) Networks: Synthesis and Characterization

**DOI:** 10.3390/polym15132937

**Published:** 2023-07-03

**Authors:** Katalin Czifrák, Csilla Lakatos, Gabriella Szabó, Bence Vadkerti, Lajos Daróczi, Miklós Zsuga, Sándor Kéki

**Affiliations:** 1Department of Applied Chemistry, University of Debrecen, Egyetem tér 1, H-4032 Debrecen, Hungary; czifrak.katalin@science.unideb.hu (K.C.); lakatoscsilla@science.unideb.hu (C.L.); szabogabi9710@gmail.com (G.S.); vadkerti.bence@science.unideb.hu (B.V.); zsuga.miklos@science.unideb.hu (M.Z.); 2Doctoral School of Chemistry, University of Debrecen, Egyetem tér 1, H-4032 Debrecen, Hungary; 3Department of Solid State Physics, University of Debrecen, Bem tér 18/b, H-4026 Debrecen, Hungary; daroczi.lajos@science.unideb.hu

**Keywords:** DGEBA-amine cooligomer, polyurethane, DGEBA-polyurethane, characterization, shape memory

## Abstract

In this work, the preparation and systematic investigation of cross-linked polyurethane-epoxy (PU-EP) polymer systems are reported. The PU-EP polymers were prepared using a reaction of isocyanate (NCO)-terminated PU-prepolymer with diglycidyl ether of bisphenol A (DGEBA)-amine cooligomer. The oligomerization of DGEBA was carried out by adding furfurylamine (FA) or ethanolamine (EA), resulting in DGEBA-amine cooligomers. For the synthesis of NCO-terminated PU-prepolymer, poly(ε-caprolactone)diol (PCD) (M_n_ = 2 kg/mol) and 1,6-hexamethylene diisocyanate (HDI) were used. The cross-linking was achieved by adding DGEBA-amine cooligomer to PU-prepolymer, in which the obtained urethane bonds, due to the presence of free hydroxil groups in the activated DGEBA, served as netpoints. During cross-linking, ethanolamine provides an additional free hydroxyl group for the formation of a new urethane bond, while furfurylamine can serve as a thermoreversible coupling element (e.g., Diels–Alder adduct). The PU-EP networks were characterized using attenuated total reflectance Fourier-transform infrared spectroscopy (ATR-FTIR), differential scanning calorimetry (DSC), dynamical mechanical analysis (DMA) and scanning electron microscopy (SEM). The DMA curves of some PU-EPs (depending on the compositions and the synthetic method) revealed a plateau-like region above the melting temperature (T_m_) of PCD, confirming the presence of a cross-linked structure. This property resulted in a shape memory (SM) behavior for these samples, which can be fine-tuned in the presence of furfurylamine through the formation of additional thermoreversible bonds (e.g., Diels–Alder adduct).

## 1. Introduction

Epoxy resins are used in many areas of our everyday lives, including electronics or structural materials in automotive and construction industries, as well as finding applications as coatings or adhesives. The wide-range application of epoxies is due to their good durability as well as their dielectric and thermal properties [1,2]. One of the most known and frequently applied epoxy resins is the diglycidyl ether of bisphenol A (DGEBA) [3,4]. DGEBA in a commercial epoxy resin system is a mixture of oligomers with a different number of repeating units (n). The physical state of these resins, depending on the molecular weight, changes from viscous liquid to brittle solid at room temperature [5].

Although the resistance of epoxies to mechanical stress is significant, they are stiff and brittle, and these undesired properties can be reduced by incorporating polyurethanes into them [6]. Furthermore, the properties of polyurethanes can be tailored to specific applications [7,8], allowing variable mechanical properties. In addition, the incorporation of biodegradable building blocks [9] (e.g., poly(ε-caprolactone)) into polymer chains can make the resulting material biocompatible and environmentally friendly [10] in addition to its softening effect [11,12] and can also develop the shape memory property [13].

Several studies reported on the cross-linking of epoxy moieties with NCO-terminated linear polyurethane [14,15,16,17,18]. This requires free OH groups formed by opening the oxirane ring to be present in the epoxy resin [19]. The reactive epoxy function, due to the electrophilicity of the carbon atoms of the ring, can be opened with nucleophiles, e.g., with amines and alcohols [2,20], etc.

The curing reaction of epoxy resins with amines also occurs via a nucleophilic attack of the amine nitrogen on the electrophilic terminal carbon of the epoxy function via the S_N_2-type mechanism. In these reactions, more reactive primary and secondary amines, catalysts and promoters are applied. It is widely accepted that in the solvent phase, primary amines show greater reactivity in the ring-opening reaction, which is also affected by the solvent’s polarity [19,21].

Furthermore, another factor influencing the reactivity of amine function is the steric hindrance [19]. Formation of a branched or linear structure is determined by the rate of the reaction of the epoxide with the primary and secondary amine hydrogen [22,23].

The thermoresponsive SMPs, which contain amorphous or semi-crystalline domains, occupy a prominent place in the family of shape-memory polymers [24]. In these systems, semi-crystalline domains play the role of switches. Polycaprolactones with different molecular weights fulfill this role due to their semi-crystalline microstructure [25]. Furthermore, by incorporating elements suitable for forming thermoreversible bonds in these polymers (e.g., Diels–Alder adduct formed from furfurylamine) [26], the shape memory effect can be tuned while achieving better mechanical properties.

In this paper, poly(ε-caprolactone)diol (PCD)-based NCO-terminated linear polyurethanes as prepolymer diisocyanates were reacted with DGEBA-amine cooligomers synthesized from the reaction of diglycidyl ether of bisphenol A (DGEBA) with monoamines (FA, EA) in solution under mild reaction conditions. The obtained cross-linked polyurethane-epoxy systems were investigated using ATR-FT-IR spectroscopy. Mechanical, thermo- and thermomechanical properties were elucidated through tensile measurements, differential scanning calorimetry (DSC) and dynamic mechanical analysis (DMA) experiments. The aim of the research was to develop polyurethane-epoxy polymer systems with suitable mechanical and thermomechanical properties for technical use and with the shape memory property.

## 2. Experimental Procedure

### 2.1. Materials

Poly(ε-caprolactone)diol (PCD) (M_n_ = 2 kg/mol) from Sigma-Aldrich Chemical Co. (Darmstadt, Germany) was used as the diol component; 1,6-Hexamethylene diisocyanate (HDI), Furfurylamine (FA, >99%) and catalyst Tin(II) 2-ethylhexanoate (M = 405.12 g/mol, 92.5–100%) were all of reagent grade and purchased from Sigma-Aldrich Chemical Co. (Darmstadt, Germany). Ethanolamine (EA, 99%) was purchased from Loba Feinchemie Gmbh (Fischamend, Austria). Bisphenol A diglycidyl ether type EP oligomer (DGEBA, epoxy equivalent weight: 182–192 g/mol) was provided by Alvinplast (Budapest, Hungary). Toluene from Molar Chemicals Ltd. (Halásztelek, Hungary) was distilled over P_2_O_5_ and stored on sodium wire until use.

### 2.2. Syntheses of PU-EPs

A typical procedure for synthesis of epoxy PUs:

Syntheses of PU-prepolymers: for the syntheses, a preheated 100 mL round-bottom, three-necked flask equipped with a magnetic stirrer, argon inlet, reflux condenser and CaCl_2_ tube was used. In this vessel, 5 g (2.5 mmol) of PCD (2 kg/mol) was dissolved in 30 mL of dry toluene. The temperature was kept at 80 °C with a silicon oil bath. To this solution, 0.81 mL (5.0 mmol, 2 equiv.) of 1,6-hexamethylene diisocyanate (HDI) was added in the presence of 2 mol% (0.02 g, 0.05 mmol) of Tin(II) 2-ethylhexanoate as a catalyst, and the mixture was stirred at 80 °C under continuous nitrogen flow for 2 h.

Syntheses of DGEBA-amine cooligomers (Figure 1): 0.9 g (2.5 mmol, 1 equiv.) of DGEBA and amine (EA or FA) 0.1–0.3 equiv. was dissolved in dry toluene (5 mL) and stirred at 80 °C for 2 h under an argon atmosphere. The reaction mixture of DGEBA-amine cooligomers was investigated with MALDI-TOF MS (see Figure 1 and Figure 1).

Cross-linking: The combined mixture of PU prepolymer and DGEBA solutions was stirred for another 2 h at the same temperature (80 °C). Then the obtained reaction mixtures were poured into Teflon plates and were cured at 40 °C for another 12 h until a constant mass was reached (e.g., the theoretical mass: 6.76 g, after drying: 6.73 g for PU-EP 1) to give an elastic polymer film (Table 1).

### 2.3. Characterization

The Matrix-Assisted Laser Desorption/Ionization Time-of-Flight Mass Spectrometry (MALDI-TOF MS) measurements were performed with a Bruker Autoflex Speed mass spectrometer equipped with a time-of-flight/time-of-flight (TOF/TOF) mass analyzer (Bruker Daltoniks, Bremen, Germany). In all cases, 19 kV acceleration voltage was used in the positive ion mode. To obtain appreciable resolution and mass accuracy, ions were detected in reflectron and linear mode, where 21 kV and 9.55 kV were applied as reflector voltages. A solid-phase laser (355 nm, ≥100 μJ/pulse) operating at 500 Hz was applied to produce laser desorption and 5000 shots were summed. The MALDI-TOF MS spectra were externally calibrated with standard polyethylene glycol (M_n_ = 1540 g/mol). Samples for MALDI-TOF MS were prepared with 2,5-dihydroxy benzoic acid (DHB) matrix dissolved in THF at a concentration of 20 mg/mL. The samples and the sodium trifluoroacetate used as an ionizing agent were also dissolved in THF at a concentration of 10 mg/mL and 5 mg/mL, respectively. The mixing ratio was 10/2/1 (matrix/sample/cationizing agent). A volume of 0.25 μL of the solution was deposited onto a metal sample plate and allowed to air-dry.

Attenuated total reflectance (ATR) Fourier-transform infrared (AT-FTIR) spectra were recorded on a Perkin Elmer Instruments (Waltham, MA, USA) Spectrum Two FTIR spectrometer equipped with a diamond Universal ATR Sampling Accessory. The average film thickness of the specimens was ca. 0.5 mm. Eight scans were taken for each sample. The spectra were evaluated using the Spectrum ES 5.0 program.

For investigation of surface morphology, scanning electron microscopy (SEM) was adopted. SEM pictures were taken from the surface of selected specimens using a Hitachi S-4800 microscope (Tokyo, Japan) equipped with a Bruker energy dispersive X-ray spectrometer. For microscopic examinations, 1 cm × 1 cm samples with an average thickness of 0.5 mm were excised. The surface of the specimens was covered with a 30 nm conductive gold layer. The SEM observations were performed at 10 kV accelerating voltage in secondary electron mode.

For the determinations of the degree of swelling (Q), gel content (G) and cross-link density (ν_e_), samples (dimension: 10 × 10 × ~0.5 mm) were swollen in toluene (20 mL) at 22 °C (295 K) in a closed bottle for 48 h. The degree of swelling (Q), the gel content (G) and cross-link density were calculated with Equations (1)–(3) [27,28].
(1)$$Q=1+\frac{\rho_{s}}{\rho_{p}}\left(\frac{m_{2}}{m_{3}}-1\right)$$
(2)$$G\left(\%\right)=\frac{m_{3}}{m_{1}}\cdot 100$$
where ρ_s_ and ρ_p_ are the densities of the solvent (toluene, ρ: 0.8669 g/cm^3^) and the PU polymer, respectively.

The cross-link density (ν_e_) was calculated based on the swelling results using the Flory–Rehner Equation (3) [29]:(3)$$\nu_{e}=\frac{-\left[\ln\left(1-v_{1}\right){+v}_{1}+\chi \cdot v_{1}{}^{2}\right]}{V_{\mathrm{ms}}\cdot \;\left(v_{1}{}^{1/3}-\frac{v_{1}}{2}\right)}$$
where ν_e_ = cross-link density, V_ms_ is the molar volume of the solvent (1.06 × 10^−4^ m^3^/mol), χ is the polymer–solvent interaction parameter (calculated to be 0.228 at 299 K) and ρ_p_ is the density of the polymer [30].

For the tensile test, a computer-controlled Instron 3366 (Instron, Norwood, MA, USA) type tensile testing machine equipped with a 100 kN load cell was used. Tensile tests were carried out according to the EN ISO 527-1 standard. At least five dumbbell specimens were cut (clamped length 60 mm) and measured based on the EN ISO 527-3 standard and tensile loaded at a crosshead speed of 50 mm/min. The thickness of the specimens was around ~0.5 mm. The measurement data of the five specimens were evaluated with Instron Bluehill Universal V 4.05 (2017) software.

The thermal properties of the synthesized PU-EPs were carried out using differential scanning calorimetry (DSC). DSC tests were performed in DSC 3 power compensation equipment operating at a 10 °C/min heating rate. Nitrogen flushing was used as protective atmosphere. During DSC measurements, heat/cool cycles were carried out. For the heating cycle, the temperature was raised from −70 °C to 220 °C by a 10 °C/min heating rate. For the cooling cycle, the sample cooled down from 220 °C to −70 °C using the same heating rate. The weight percentage of the crystalline PCD unit (C_r_) was calculated with Equation (4) [31]:(4)$$C_{r\;}\left(\%\right)=\frac{∆H_{m}}{\chi_{\mathrm{PCD}}\cdot \;∆H_{m}{}^{0}}\;\cdot 100$$
where ∆H_m_ is the heat of fusion of the investigated PU, χ_A_ is the weight fraction of PCD in the corresponding PU-EP composites and ∆H_m_^0^ is the heat of fusion of the pure 100% crystalline PCD [32].

Dynamic mechanical analysis (DMA) testing of the PU-EP samples was carried out with METRAVIB (Limonest, France) DMA 25 instruments. DMA traces were monitored in tension mode (dimension of the specimens: length: 30 mm, clamped length: 19 mm, width: 15 mm, thickness: ca. 0.5 mm). A dynamic displacement of 0.1 mm was applied at frequencies of 10 Hz. The temperature was varied between −30 °C and 150 °C with a heating rate of 3 °C/min.

Shape memory experiments were evaluated in tensile mode using the Q800 DMA device of TA. The specimens (clamped length × width × thickness = ca. 12 mm × 7 mm × 0.5 mm) were stretched after 10 min holding at 60 °C at a strain rate of 15%/min to 30% strain, followed by quickly cooling the specimen to 0 °C. The stress was then released, and the shape fixity (R_f_) was determined. Shape recovery (R_r_) was measured at 0.1 N loading of the specimens (quasi-free recovery) by reheating the specimens at a 1 °C/min heating rate from 0 °C to 60 °C and holding there for 20 min. The shape fixity (R_f_) and shape recovery ratios (R_r_) are defined by Equations (5) and (6).
(5)$$R_{f}\left(\%\right)=\frac{l_{d}-l_{0}}{l_{30\%}-l_{0}}\cdot 100$$
(6)$$R_{r}\left(\%\right)=\frac{l_{d}-l_{f}}{l_{d}-l_{0}}\cdot 100$$
where l_d_ is the sample length after removal of the tensile load during shape fixing at 0 °C, l_0_ is the clamped length of the sample at 20 °C, l_30%_ is the length after tensile stretching for 30% and l_f_ is the final recovered length of the stretched specimen.

## 3. Results and Discussion

### 3.1. Oligomerization of the DGEBA Resin with Amines

The DGEBA-amine cooligomers were investigated with MALDI-TOF MS (Figure 1 and Appendix A). The number-average molecular weights are given in Table 2. As Figure 1 illustrates, the obtained DGEBA-EA cooligomer consists of mostly two DGEBA units coupled by one amine unit after 2 h of reaction time. Smaller amounts of longer chain epoxy–amine cooligomers (three DGEBA and two amine units (q)) were also formed. In addition to ethanolamine, furfurylamine also gave a reaction mixture of a similar composition (Appendix A). By increasing the amount of amine, the length of the epoxy-cooligomer chain was growing after 2 h, which, in the case of a composition of 1:0.3, included nine DGEBA and eight ethanolamine units, while six furfurylamines were connected to the five DGEBA units (see Appendix A).

The formation of the epoxy–amine cooligomer is influenced by the structure of both the amine and the epoxy moieties. During the synthesis of cooligomers, the reactive epoxy component was the diglycidyl ether bisphenol A (DGEBA), having epoxy chain-ends in the beta position with an oxygen heteroatom. The reactivity of the amine unit can also be increased if there is a heteroatom in the amine structure that, in addition to its negative inductive effect, is able to form H-bonds with the epoxy group, thus positioning it properly for ring opening. During the cooligomerization, the reaction of a primary amine with an epoxy group leads to the formation of a secondary amine, which can transform into a tertiary amine with another epoxy group without active hydrogen. The resulting tertiary amine can act as a catalyst during cooligomerization process [33].

### 3.2. Synthesis and Characterization of Polyurethane-Epoxy (PU-EP) Copolymers

Our aim was the synthesis of cross-linked PU-EP systems through the reaction of NCO-terminated PU-prepolymer with an epoxy–amine cooligomer. The oligomerization of DGEBA was carried out by adding 0.1–0.3 equivalent of monofunctional primer amine to the solution of DGEBA to yield DGEBA-amine cooligomers with different chain lengths (Figure 1).

The cross-linked structure in these systems can be obtained by grafting a urethane bond between the present free OH groups of the DGEBA cooligomer and the NCO groups of PU-prepolymers (Figure 2). On the other hand, in the presence of furanyl groups, i.e., the moieties of the furfurylamine comonomer, the shape memory phenomenon can be fine-tuned through an additional click reaction, e.g., through the formation of a thermoreversible Diels–Alder adduct. The amount of amine to that of DGEBA affects the size of cooligomer chains and hence, the stiffness of the epoxy unit in polyurethane. In the following, we attempt to interpret this effect by analyzing the data obtained during chemical, mechanical and thermomechanical testing of the prepared samples.

### 3.3. Infrared Spectroscopy

In order to investigate the chemical structure of the samples, IR spectra were taken from the PU-EPs 1–6 polyurethane-epoxy copolymers after cross-linking and solidification (Figure 2 and Appendix A) [16]. The broad absorption band with low intensity between 3363 and 3335 cm^−1^ belongs to the urethane –NH bands. The –CH_2_ vibrations appear as a double band at around 2940 and 2860 cm^−1^, indicating the presence of a large number of CH_2_ groups in these polymers. The lack of the absorption band around 2230 cm^−1^ supports the complete reaction of the NCO functional groups. The –C=O appears between 1726 and 1721 cm^−1^ in the PCD-based PU-EPs. A peak with small intensity appears as a shoulder at around 1680 cm^−1^, indicating the presence of the H-bonded –C=O group.

The peaks of =C–O and –C–O–C– bonds appear between 1242 and 1233 and 1189 and 1155 cm^−1^ in PU-EP copolymers. The very small peak around 915 cm^−1^ indicates the low amount of unreacted epoxy groups.

Significant differences with the increasing amount of amine cannot be seen by comparing the IR spectra of all samples due to the large amount of caprolactone unit.

### 3.4. Swelling Experiments

To characterize the obtained network, among others, swelling experiments were carried out. The swelling degree (Q), values of the gel content (G) and cross-link densities obtained are summarized in Table 3. The values of the swelling degree change between 1.9 and 3.7, while the gel contents of PU-EP samples vary from 67.8% to 88.7% depending on the compositions.

The cross-link density values were obtained in the order of 10^−3^ mol/cm^3^ and 10^−4^ mol/cm^3^. Furthermore, it can be noticed that in the case of sample PU-EPs 1–3, the cross-link densities show a decrease with increasing molar ratios of the reactants toward the equimolar quantities.

These experimental facts are a clear indication that the synthesis of both DGEBA-amine cooligomers and the grafting reactions yielding networks are step reactions.

### 3.5. Morphology of PU-EPs Using SEM

The SEM images of PU-EP samples 1–6 are shown in Figure 3.

Figure 3 and Appendix A show that a small extent of micro phase separation of PU-EP 1 and PU-EP 4 takes place. These findings support the change of cross-link densities (see Table 1 and Table 3), i.e., the highest values of cross-link densities were found to be at the lowest molar ratios of reactants.

### 3.6. Mechanical Properties of PU-EPs

The stress–strain curves of PU-EPs 1–6 are shown in Figure 4.

From the obtained experimental results, it is obvious that the presence of a smaller amount of amine (0.1 eq.) gives the best mechanical properties in the case of both amines (Table 1 and Table 3). However, the increasing amount of amine decreases the flexibility and toughness of the samples, as evidenced by decreasing elongation and tensile strength values. This is due to the formation of longer epoxy–amine cooligomer chains by increasing the amount of amine, which results in a looser cross-linked structure.

It can be seen from the results of the tensile measurements that grafting with epoxy–amine cooligomers of different lengths affects the mechanical properties of networks obtained (Figure 4 and Table 4). It is also to be noted that, with the exception of the PU-EP 6 sample, the stiffening effect of the epoxy block is not significant due to the softening effect of the PCD.

### 3.7. Mathematical Modeling of Stress–Strain Curves

The Standard Linear Solid (SLS) model is often used to model the behavior of viscoelastic materials, as it is able to take creep and stress relaxation into account. Therefore, the SLS model was used to describe the stress-relative strain curves, where Equation (7) and its extended form (Equation (8)) were used to render the strain-hardening effect in the case of our samples.
(7)$$\frac{d\sigma }{d\varepsilon }={\left(\frac{d\varepsilon }{\mathrm{dt}}a_{1}\right)}^{-1}\left[a_{2}\left(\varepsilon \right)\varepsilon +{\;a}_{3}\frac{d\varepsilon }{\mathrm{dt}}-\sigma \right]$$
(8)$${(a}_{2}\left(\varepsilon \right){=a}_{2}+\alpha \;(\varepsilon -\varepsilon_{L}{)}^{\beta }{\;(\mathrm{if}\;\varepsilon \;>\;\varepsilon }_{L})$$
where σ and ε are the stress and relative strain, respectively; a_1_, a_2_, a_3_ and σ are the parameters to be determined and ε_L_ is the “critical” strain at which the strain-hardening effect occurs.

In these calculations, β = 1 was applied, and the numerically integrated forms of Equations (7) and (8) were fitted to the experimental stress–strain curves to obtain the expected parameters. This theoretical calculation, including Equations (7) and (8), proved to be successfully applied in the case of many previously produced polyurethanes [34,35,36,37]. As presented in Figure 5, the curves obtained as a result of the theoretical calculation match the experimental data quite well, supporting the applicability of the SLS model for these polymer systems.

### 3.8. Thermal and Thermomechanical Behavior

The thermal behavior of PU-EPs 1–6 was investigated using DSC. In these samples, the PCD segment was responsible for the crystallization and worked as a “switch” upon melting (Table 1). As Figure 6 and Table 5 show, the glass transition temperatures (T_g_) of PCD-based samples appeared in the range of −46.7 to −50.6 °C, while T_m_ values changed from 18 to 23 °C. The T_g_ values were shifted to higher temperatures compared to the T_g_ of the pure poly(ε-caprolactone)diol (−64 °C, see ref. [32]) due to the presence of cross-links, which decrease the chain mobility and thus increase the values of T_g_.

The degree of crystallinity values was calculated based on PCD weight content in each of the PUs using the reference ∆H_m_^0^ values of PCD [32]. The degree of crystallinity rate was below 25% for all samples, but it was particularly low for sample PU-EPs 5 and 6.

The crystallization of polycaprolactone chains is influenced by the cross-linked structure [38]. In these polymers, the formed urethane bonds can serve as netpoints during grafting. Thus, the degree of crystallinity (DC_r_) decreases with the increasing formation of the cross-linked structure. Hence, this finding can be explained by the presence of a cross-linked structure, which reduces the final degree of crystallization [39].

In addition to the stress storage capacity, the presence of the cross-linked structure can also be verified from the storage modulus curves.

The storage modulus as a function of temperature is plotted in Figure 7 and Appendix A.

The existence of a cross-linked structure is supported further by the presence of rubbery plateaus appearing on these curves. The cross-link density in the plateau region (ν_d_) can be calculated according to Equation (9) for samples PU-EPs 1, 2 and 4, 5 at 323 K (see Table 6).
(9)$$\nu_{d}=\frac{E^{\prime }}{3\;R\;T}$$
where E′ and R are the storage modulus at temperature T in Kelvin and the universal gas constant, respectively.

The obtained cross-link densities are in the order of 10^−3^, and a decrease is observed with the growing amine–epoxy cooligomer chain. The storage modulus curves for samples 3 and 6 are shown in Appendix A. The formation of the plateau cannot be seen on these curves. All of this is in line with the results obtained for the swelling and tensile tests, i.e., a lower cross-link density is associated with a higher amount of amine in the DGEBA-amine cooligomer.

Based on the previous results, shape memory experiments were performed for the sample PU-EPs 1 and 4 (Figure 8).

The values of shape fixity for PU-EPs 1 and PU-EP 4 (R_f_) are 60.4% and 62.2%, while the shape recovery ratios (R_r_) are 64% and 55%, respectively. The moderate shape fixity ratios are the result of the mostly amorphous structure of the PCD unit; through its melting, it fulfills the role of the switching element. In the case of PCD, due to the relatively low melting point, shape fixations were performed at 0 °C during the measurement. The value of the shape recovery ratio is also influenced by the epoxy–amine cooligomer, which is slightly higher in sample PU-EP 4. The moderate shape recovery ratio is due to the presence of the cooligomer chains, which significantly reduces the degree of crystallinity and thereby increases the proportion of the amorphous segment in the polymer microstructure.

## 4. Conclusions

The synthesis of epoxy–amine cooligomers was carried out through the reaction of diglycidyl ether of bisphenol A (DGEBA) and amines (ethanolamine, furfuryalmine), where the amount of the amine was systematically increased. The formation of cooligomers was investigated using MALDI-TOF MS. In order to obtain cross-linked PU-EP materials, the amine–epoxy cooligomers were reacted with PCD-based PU prepolymer. The network structure formed from the epoxy–amine cooligomer chain length was investigated using ATR-FTIR, swelling, and mechanical and thermomechanical tests used to evaluate the formed cross-linked polymer systems. Among the synthesized polyurethane-epoxy systems, the samples PU-EP 1, 2 and 4, 5 had a low degree of crystallinity (DC_r_ was not more than 25%) and exhibited excellent mechanical properties. In addition, the shape memory cycle test for samples PU-EP 1 and 4 revealed that they bore the shape memory property, although the shape fixity and shape recovery values were relatively moderate in the presence of PCD as a switching element.

## Data Availability

Not applicable.

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
