# Peer review of "Bisphenol A Diglycidyl Ether-Primary Amine Cooligomer-poly(ε-caprolactone) Networks: Synthesis and Characterization"

_polymers, 2023, doi:10.3390/polym15132937_

Round 1
Reviewer 1 Report
In this work, the authors report the crosslinked polyurethane-epoxy (PU-EP) polymer systems by reaction of isocyanate (NCO) terminated PU-prepolymer with diglycidyl ether of bisphenol A (DGEBA)-amine cooligomer, and investigate structure characterization, swelling and mechanical properties, thermal and thermomechanical behavior, and shape memory. The manuscript is very well-organized, the experimental work is well conducted. I would like to recommend this manuscript for publication after minor revision.
Some comments to the Authors:
1. GPC curves or molecular weight of DGEBA-amine cooligomers and PU-prepolymer should be supplied.
2. Thermal stability or TGA curves of PU-EP polymer networks should be investigated or supplied. 3. Tg of PCD based samples appeared in the range of -46.7 – -50.6 °C, the authors should explain lower Tg. 4. The reason of low shape memory recovery rate should be explained.
Author Response
June 30, 2023
Dear Reviewer,
Thank you for reviewing our manuscript entitled „Novel shape-memory DGEBA primary amine cooligomers of poly(e-caprolactone) networks: Synthesis and characterization”. Our answers for your comments and questions are as follows:
GPC curves or molecular weight of DGEBA-amine cooligomers and PU-prepolymer should be supplied.
Reply: Thank you for your suggestion. The Mn values calculated based on the MALDI-TOF MS spectra of DGEBA-amine cooligomers are now given in Table 2. The MALDI-TOF MS spectra are inserted into the Supplementary Material as Fig. S1-Fig. S7.
Thermal stability or TGA curves of PU-EP polymer networks should be investigated or supplied.
Reply: We agree with our reviewer that testing the thermal stability of epoxy-polyurethane samples may be important. However, the thermal stability of the polymer samples was not investigated, as we did not have access to TGA measurements.
Tg of PCD based samples appeared in the range of -46.7 – -50.6 °C, the authors should explain lower Tg.
Reply: Thank you for your note. The main component in our polymers is poly(e-caprolactone)diol (PCD), whose glass transition temperature is around -64 °C (see ref. S. McCreath, P. Boinard, E. Boinard, P. Gritter, J.J. Liggata. International Journal of Adhesion and Adhesives, 2018, 86, 94-97). The presence of crosslinks decrease the chain mobility thus increaseasing the Tg value of PCD.
The reason of low shape memory recovery rate should be explained.
Reply: The reason for the low shape recovery ratio stems from the fact that poly(e-caprolactone)diol (PCD), whose melting point, glass transition temperature and degree of crystallinity are not very high, plays the role of the switch segment in our polymers. On the other hand, the presence of the epoxy-amine cooligomer causes a further decrease in the degree of crystallinity and the melting point of the PCD unit based on DSC measurements.
The following sentence was inserted into the text: “The moderate shape recovery ratio is due, most probably, to the presence of the cooligomer chains, which significantly reduce the degree of crystallinity and thereby increase the pro-portion of the amorphous segment in the polymer microstructure.”
Yours sincerely,
Sándor Kéki

Reviewer 2 Report
I have reviewed this manuscript. However, I found the novelty of chemistry in this manuscript is not high. The hydroxyl in the ethanol amine should also involve the curing reaction (Scheme 2). Why do the authors select the furfuryl ammine as a reactant? The design is not explained. Therefore, I cannot agree to publish this manuscript.
What is the main question trying to address by the research?
Epoxy thermosets are mechanically hard materials, the authors try to bring softness to the epoxy network, which is widely researched already.
In introduction they mentioned that they have used poly(ε-caprolactone) to bring bio-degradability aspect to the epoxy network. Which is current trend. I noted that the authors mix HDI with poly(ε-caprolactone) and say that they would like to bring bio-degradability, which is contrary to reality as the BPA is known for its endocrine disruptive nature when leaked into environment. How authors can explain/justify bio-degradability here?
Do you consider the topic original or relevant in the field? Does it address a specific gap in the field?
Problem 1. The main concept of this manuscript is to bring softness to the hard epoxy thermosets (elastomerization through introduction of PU network). The authors succeed in bringing the softness into the epoxy system.
Problem 2. Bio-degradability aspect was not studied. As authors mentioned that they have used PCD to bring bio-degradability into the polymeric network, but I did not notice any scientific discussion on it.
My suggestion: Chemical degradation is worth in their system.
What does it add to the subject area compared with other published materials?
It adds to the DGEBA derived elastomers, the authors studied their elastomeric properties. I did not find something really new through!
What Specific Improvements should the authors consider regarding the methodology? What further controls should be considered?
Since epoxy networks are hard to recycle, the current researchers focused not only a good performance but also a good recyclability, so that end-of-life polymeric networks would not get wasted in the environment (just keep adding to the pollution).
Are the conclusions consistent with the evidence and arguments presented and do they address the main question posed?
Conclusions are consistent with respective to results obtained. One of the main questions posed was recyclability, the authors did not explore it.
Are the references appropriate?
Yes.
Please include any additional comments on the tables and figures?
The figure resolution and quality of the figures can be improved.
Author Response
June 30, 2023
Dear Reviewer,
Thank you for reviewing our manuscript entitled „Novel shape-memory DGEBA primary amine cooligomers of poly(e-caprolactone) networks: Synthesis and characterization”. Our answers for your comments and questions are as follows:
The hydroxyl in the ethanol amine should also involve the curing reaction (Scheme 2)
Reply: Thank you for this note. We accepted the reviewer suggestion and Scheme 2 was modified accordingly.
Why do the authors select the furfuryl amine as a reactant? The design is not explained.
Reply: The aim of the work was to create epoxy-polyurethane systems with good mechanical and shape memory properties. During the syntheses, we worked with monoamines to create simpler structures. In these structures the ethanolamine serves an additional free OH group in the cross-linking reaction. On the other hand, furfurylamine, as a thermo-reversible coupling element, enables further fine-tuning of the shape memory property, e.g. through the formation of thermo-reversible Diels-Alder adduct. These concepts are mentioned in the abstract and in the Introduction.
Yours sincerely,
Sándor Kéki

Reviewer 3 Report
This manuscript studies the preparation and systematic investigation of crosslinked polyurethane-epoxy (PU-EP) polymer systems. The paper is well-organized, and the title is practical and attractive. In addition, the following should also be considered before publishing.
Innovation, the purpose of the research, the tests performed, and the results and achievements should be included in the abstract quantitatively and qualitatively. So, the abstract should be written better and needs revisions. The article needs general writing and grammar editing.
The introduction is brief, superficial, and incomplete. The number of used and reviewed references is very small. Also, the paragraphs presented are primarily general and general information. At the end of the introduction, a suitable summary of the importance of the present issue should be provided.
Use the following resources to deepen the introduction. Toughening PVC with Biocompatible PCL Softeners for Supreme Mechanical Properties, Morphology, Shape Memory Effects, and FFF Printability. 4D Printing‐Encapsulated Polycaprolactone–Thermoplastic Polyurethane with High Shape Memory Performances.
How has the reproducibility of these results been checked? The tensile test standard, displacement measurement method and accuracy, and the number of repetitions of the results should also be mentioned.
The results section is well organized and categorized. But some parts report the results, which require corrections and deepening the analysis and discussion. In the conclusion section, a summary of the purpose of the research, innovation, and research method should be presented before presenting the highlights.
No comment.
Author Response
June 30, 2023
Dear Reviewer,
Thank you for reviewing our manuscript entitled „Novel shape-memory DGEBA primary amine cooligomers of poly(e-caprolactone) networks: Synthesis and characterization”. Our answers for your comments and questions are as follows:
This manuscript studies the preparation and systematic investigation of crosslinked polyurethane-epoxy (PU-EP) polymer systems. The paper is well-organized, and the title is practical and attractive. In addition, the following should also be considered before publishing.
Innovation, the purpose of the research, the tests performed, and the results and achievements should be included in the abstract quantitatively and qualitatively. So, the abstract should be written better and needs revisions. The article needs general writing and grammar editing.
Reply: Thank you for your remark. The abstract has been modified accordingly. The grammar was also corrected.
The introduction is brief, superficial, and incomplete. The number of used and reviewed references is very small. Also, the paragraphs presented are primarily general and general information. At the end of the introduction, a suitable summary of the importance of the present issue should be provided.
Reply: Thank you for your note. The introduction was modified as follows:
„The thermo-responsive SMPs, which contain amorphous or semi-crystalline domains, occupy a prominent place in the family of shape-memory polymers [24]. In these systems, semi-crystalline domains play the role of switches. Poly-caprolactones with different molecular weights fulfill this role due to their semi-crystalline microstructure [25]. Furthermore, by incorporating elements suitable for forming thermo-reversible bonds in these polymers (e.g. Diels-Alder adduct formed from furfurylamine) [26], the shape memory effect can be tuned while achieving better mechanical properties.”
„The aim of the research was to develop polyurethane-epoxy polymer systems with suitable mechanical and thermomechanical properties for technical use and with shape memory property.”
Use the following resources to deepen the introduction. Toughening PVC with Biocompatible PCL Softeners for Supreme Mechanical Properties, Morphology, Shape Memory Effects, and FFF Printability. 4D Printing‐Encapsulated Polycaprolactone–Thermoplastic Polyurethane with High Shape Memory Performances.
Reply: Thank you for your remark. The proposed subject areas in the introduction were mentioned.
See the following references inserted into the introduction:
„11. Rahmatabadi, D.; Aberoumand, M.; Soltanmohammadi, K.; Soleyman, E.; Ghasemi, I.; Baniassadi, M.; Abrinia, K.; Bodaghi, M.; Baghani, M. Toughening PVC with Biocompatible PCL Softeners for Supreme Mechanical Properties, Morphology, Shape Memory Effects, and FFF Printability. Macromol. Mater. Eng. 2023, 2300114.
- Mina Hernandez, J.H. Effect of the Incorporation of Polycaprolactone (PCL) on the Retrogradation of Binary Blends with Cassava Thermoplastic Starch (TPS). Polymers 2021, 13, 38.
- Lendlein, A.; Schmidt, M.A.; Langer, R. AB-polymer networks based on oligo((-caprolactone) segments showing shape-memory properties. Proceeding of the National Academy of Sciences of the United States of America 2001, 98, 842–847.
- Xia, Y.; He, Y.; Zhang, F.; Liu, Y.; Leng, J. A Review of Shape Memory Polymers and Composites: Mechanisms, Materials, and Applications. Adv.Mater. 2021, 33, 2000713.
- Chung, T.; Romo-Uribe, A.; Mather, P.T. Two-Way Reversible Shape Memory in a Semicrystalline Network. Macromolecules 2008, 41, 184–192.
- Jiang, Y.; Hadjichristidis, N. Diels-Alder Polymer Networks with Temperature-Reversible Cross-Linking-Induced Emission. Angew. Chem. Int. Ed. 2021, 60, 331–337.”
How has the reproducibility of these results been checked? The tensile test standard, displacement measurement method and accuracy, and the number of repetitions of the results should also be mentioned.
Reply: Thank you for your note. The tensile tests were carried out according to the EN ISO 527-1 standard. At least five dumbbell specimens were cut and measured based on EN ISO 527-3 standard. The thickness of the specimens was around ~0.5 mm. The measurement data of the five specimens were evaluated by Instron Bluehill Universal V 4.05 (2017) software.
The results section is well organized and categorized. But some parts report the results, which require corrections and deepening the analysis and discussion. In the conclusion section, a summary of the purpose of the research, innovation, and research method should be presented before presenting the highlights.
Reply: Thank you for your remark. The mentioned parts of the paper were modified.
Yours sincerely,
Sándor Kéki

Round 2
Reviewer 2 Report
The authors have responded to my questions. I do not have other questions.